# Shoulder Physiological Offset Parameters in Asian Populations—A Magnetic Resonance Imaging Study

**DOI:** 10.3390/diagnostics15020146

**Published:** 2025-01-09

**Authors:** Hung-Yi Huang, Meng-Hao Lin, Chu-Hsiang Hsu, Liang-Tseng Kuo

**Affiliations:** 1Department of Education, Chang Gung Memorial Hospital, Chiayi 613016, Taiwan; davy30331@gmail.com; 2Department of Orthopaedic Surgery, Chang Gung Memorial Hospital, Yunlin 638502, Taiwan; skdan1108@gmail.com; 3Department of Radiology, Chang Gung Memorial Hospital, Chiayi 613016, Taiwan; hn193309482@gmail.com; 4Department of Sports Medicine, Landseed International Hospital, Taoyuan 324609, Taiwan

**Keywords:** humeral offset, glenoidal offset, MRI, shoulder morphology, Asian

## Abstract

**Background/Objectives**: Physical shoulder offset parameters (SOPs) play important roles in the diagnosis and treatment of shoulder diseases. However, there is little research analyzing SOPs in healthy shoulders using cross-sectional MRI images, especially in Asians. Therefore, this study aimed to establish physiological reference values of shoulder parameters for Asian populations. **Methods**: This was a retrospective imaging study using MRI images of the shoulder joints of 500 patients (mean age: 55.9 ± 14.0 years). We measured the following SOPs of the normal joint: HO, GO, lateral glenoidal humeral offset (LGHO), humeral shaft axis offset (HAO), and cortical offset (CO). In addition, the offset parameters were examined for associations with age, gender, side, and osteoarthritis. **Results**: The mean HO was 22.9 (±2.4) mm, the mean GO was 62.3 (±6.6) mm, the mean LGHO was 48.9 (±4.2) mm, the mean HAO was 25.2 (±2.8) mm, and the mean CO was 15.7 (±2.7) mm. Male patients exhibited significantly higher values across all SOPs compared to female patients. In addition, there was a significantly lower mean value for HAO in left shoulders (HAO: 24.7± 2.8 mm vs. 25.5 ± 2.8 mm, *p* = 0.011). There was a negatively significant correlation between age and all SOPs. No significant difference in mean values was noted between shoulders with osteoarthritis and non-osteoarthritis in any SOPs. **Conclusions**: Significant gender- and age-specific differences were noted for all measured SOPs. In addition, right shoulders did not show higher mean SOP values than left shoulders, except for HAO, suggesting that the contralateral joint is a reliable reference for surgical planning. These findings should be considered in shoulder surgery planning.

## 1. Introduction

Physical shoulder offset parameters (SOPs) play important roles in the diagnosis and treatment of shoulder diseases [1,2,3]. Five major SOPs are essential in shoulder biomechanics and surgical planning, including humeral offset (HO), glenoidal offset (GO), lateral glenoid humeral offset (LGHO), humeral shaft axis offset (HAO), and cortical offset (CO) [4]. HO and GO influence joint stability, alignment, and loading, particularly in arthroplasty. LGHO assesses overall lateralization, maintaining rotator cuff and deltoid tension. HAO measures the humeral head’s position relative to the shaft axis, aiding implant design. CO relates the humeral cortex to the head center, serving as a predictor for function and alignment. These metrics guide the understanding and management of shoulder mechanics.

In recent years, shoulder arthroplasty has remained one of the significant orthopaedic surgeries for patients suffering from osteoarthritis and inflammatory arthritis. Several clinical studies have shown remarkable enhancements in patient-reported outcomes, pain reduction, function improvement, and overall satisfaction following shoulder arthroplasty procedures [5,6]. Similarly, humeral head-preserving surgery is essential for patients who suffer from humerus fractures with bone fragments shifting out of the normal position. Especially, younger patients who tend to experience high-velocity trauma fractures are suggested to undergo surgical intervention since the severe displacement of the humeral head is commonly noted in those situations [7,8]. To restore optimal physiological movement patterns and relieve pain in the glenohumeral joint, the surgeon should place the appropriate implant or reposition the dislocated bone in the most favorable orientation, whether in joint-preserving or joint-replacing therapies [7]. To achieve this goal, acceptable ranges of physiological shoulder offset parameters should be determined. This can be beneficial not only for preoperative diagnosis but also for assisting surgeons in formulating future treatment plans for patients.

In different surgical techniques, changes in SOPs can lead to varying effects on postoperative function and radiographic results [9,10,11]. HO and GO are crucial parameters, influencing the outcomes of arthroplasty and osteosynthesis. Previous studies have revealed that, in reverse total shoulder arthroplasty (RTSA), increasing the GO individually may minimize the possibility of scapular notching and bony impingement after surgery. At the same time, a higher GO can improve shoulder stability and joint motion, although at the expense of increasing the risk of baseplate loosening and acromial fractures and decreasing deltoid muscle efficiency [11,12,13]. Furthermore, increasing the HO in isolation may help decrease the occurrence of scapular notching, enhance rotational movement, and improve deltoid efficiency, but it may lower joint stability [1,11,12,13]. Additionally, when it comes to LGHO, higher values can result in better joint motion and more favorable clinical outcomes [11]. LGHO can also be a predictor for plate-and-screw construct failure in proximal humerus proximal fixation [14].

The recent literature suggests that evaluating physiological shoulder morphology fully using two-dimensional radiograph images can be challenging [15,16]. In terms of its structure, the shoulder joint is a ball-and-socket synovial joint, which is classified as a diarthrodial, multiaxial joint functionally [17]. Due to the above reasons, three-dimensional imaging may provide more information about the interactions between the humeral head and the glenoid cavity, making it a better approach to assessing shoulder morphology [16,18]. Based on our understanding, standardized computed tomography (CT) imaging for the measurement of shoulder joint offset and inclination has been reported in previous studies [15,16]. However, there is limited literature on established measurements for the detection of SOPs in magnetic resonance imaging (MRI). CT-based methods cannot be transferred directly into MRI [4].

Thus, SOPs play an important role in affecting postoperative function and shoulder joint motion. As a consequence, preoperative planning should be carefully formulated before orthopedic surgeries such as arthroplasty and osteosynthesis, which can cause a dramatic change in SOPs. Since race-specific shoulder morphology existed, different characteristic features of the humeral morphology have been investigated concerning specific populations [19,20,21,22]. However, there is little research analyzing physiological shoulder offsets in healthy shoulders using cross-sectional MRI images [23], and no Asian-specific data have been available to date. The aims of our study are (1) to establish physiological reference values of shoulder parameters for Asian populations; and (2) to investigate side-, age-, and gender-specific variations in SOPs, and explore their relationship with OA grades. We hypothesized that SOPs would be different between patients with different demographics.

## 2. Materials and Methods

### 2.1. Patients

This retrospective study enrolled a total of 500 patients who underwent magnetic resonance imaging of the shoulder joint at Chang Gung Memorial Hospital in Chiayi between 1 January 2018 and 31 December 2022. All the patients underwent MRI examinations for routine diagnostic purposes due to their clinical discomforts. In a consistent protocol, all the patients were positioned on the MRI table using a shoulder positioning tray. The final MRI images were evaluated by one senior radiologist (JSH) and one orthopedic surgeon (HYH). The patients’ data were blinded for analysis. This study was approved by the local ethics committee (IRB No. 202401188B0) and performed per the ethical principles outlined in the Declaration of Helsinki.

### 2.2. Inclusion Criteria

All MRI images obtained by the Picture Archiving and Communicating System (PACS system) for the evaluation of shoulder joint pathologies from 1 January 2018 to 31 December 2022 were initially included. In addition, all patients who were at least 18 years of age at the time of receiving the MRI examination were included. The Kellgren and Lawrence system was applied to classify the severity of osteoarthritis [24]. In our study, patients with a Kellgren/Lawrence score of 0–2 were categorized as shoulder joint-healthy (SJH), whereas those with a score of 3–4 were classified as having shoulder joint disease (SJD). All images were initially interpreted by radiologists and used as tools for clinical diagnosis. Subsequently, each image was reinterpreted by JSH and HYH for the study.

### 2.3. Exclusion Criteria

Patients presenting with orthopedic diseases, such as fractures, bone tumors, and osteonecrosis, were deliberately excluded from the study population. Patients who have a history of previous surgical interventions, including arthroplasty, plate osteosynthesis, or any other surgical treatment involving shoulder implants, were also excluded. Likewise, patients with shoulder subluxation, irreducible dislocation, or severe rotator cuff tear were excluded. Last but not least, MRI scans of low quality or with artifacts were also eliminated from the study.

### 2.4. MRI Analysis, Parameters, and Methods of Measurement

All measurements were obtained using the PACS system and were extracted from software, Centricity Universal Viewer (RA1000, ed 2019, Buckinghamshire, UK). We categorized the osteoarthritis score of each shoulder joint based on the Kellgren/Lawrence system [20]. The definitions of HO, GO, LGHO, HAO, and CO were established according to the reference articles before the measurements were taken [4,25,26,27]. To minimize differences in each imaging plane, the MRI examination was conducted using a standardized method. For the measurement of HO, LGHO, HAO, and CO, we determined these parameters in the coronal plane, which showed the clearest image of the humeral head and shaft. For the measurement of GO, the coronal view was used to determine the center of rotation of the humeral head [4]. From the serial coronal images, the image showing the largest circular area of the humeral head was selected for localizing the center of rotation. Then, the location of the center of rotation in the axial view was determined by using measurement software to identify the corresponding point. The GO was subsequently defined as the distance between this identified point and the end of the scapular neck. The principle of measurement approach is depicted in Figure 1. All measurements of these radiographic parameters were manually performed by one radiologist (JSH) and one orthopedic surgeon (HYH). To evaluate the consistency of the measurements, the intra-observer reliability of all parameters was examined in a subset of 50 subjects through a blinded re-assessment two weeks after the first measurements. Moreover, inter-observer reliability was assessed by two observers (JSH and MHL) for 50 subjects.

### 2.5. Statistics

The mean, standard deviation, and range were reported in our study. We used the Mann–Whitney U test to analyze gender-, side-, and age-specific differences in HO, LGHO, HAO, CO, and GO. Given that signs of aging joints, such as osteophyte formation and joint space narrowing, typically become apparent after the age of 50 years [28,29], we categorized patients into two groups: those older than 50 years and those aged 50 years or younger. Similarly, the same test was also performed to compare the SJH and SJD groups for all SOPs. Using data from a previous study [23], we conducted an a priori sample size calculation using the G*Power software version 3.1 [30] to ensure 80% power to detect a statistically significant effect size of 0.5 with a two-tailed test at an alpha level of 0.05, resulting in a required sample size of 61 participants per group. A post hoc power analysis was also conducted. The Spearman correlation was applied to investigate potential associations between different SOPs and the age of patients. The reliability of intra-observer and inter-observer measurements was expressed using intraclass correlation coefficients (ICC). An ICC value >0.75 was considered as excellent reliability. A Bland–Altman analysis was also used to assess the intra-observer and inter-observer reliability [31]. The plot was constructed, with the mean of the two methods plotted on the *x*-axis and the difference between the methods on the *y*-axis. The mean difference (bias) was calculated to quantify the systematic offset between the methods, and the limits of agreement (mean difference ± 1.96 standard deviations) were determined to define the range within which 95% of the differences were expected to lie, providing a comprehensive evaluation of the assessments’ agreement. Statistical analyses were carried out using the SAS software version 9.4 (SAS Institute, Cary, NC, USA) and Microsoft Excel (Microsoft Office 2016, Redmond, WA, USA). *p* < 0.05 indicates significant differences.

## 3. Results

### 3.1. Characteristics of Patient Groups

A total of 211 males (42.2%) and 289 females (57.8%) were included in this study. A total of 200 left (40.0%) and 300 right (60.0%) shoulders were analyzed. The mean age of the patient population was 55.9 ± 14.0 years (range of 18–91 years).

### 3.2. Analysis of the Shoulder Offset Parameters

#### 3.2.1. Humeral Offset

In total, the mean value of HO was 22.9 ± 2.4 mm. The mean HO of left shoulders (*n* = 200) was 22.8 ± 2.6 mm and 23.0 ± 2.4 mm for right shoulders. In the younger patient group (18–50 years old, *n* = 155), the mean value of HO was 23.8 ± 2.5 mm, while a mean HO of 22.6 ± 2.3 mm was noted in the older patient group (>50 years old, *n* = 345). The result of the mean HO in female shoulders was 21.7 ± 2.0 mm and in male shoulders was 24.6 ± 2.0 mm.

#### 3.2.2. Glenoidal Offset

The overall mean value of GO was 62.3 ± 6.6 mm in total. The mean GO was 62.2 ± 6.6 mm for left shoulders and 62.3 ± 6.5 mm for right shoulders. In the younger patient group, the mean GO was 64.1 ± 6.3 mm, while in the older patient group, it was 61.4 ± 6.5 mm. The mean GO for female shoulders was 60.1 ± 5.9 mm, and for male shoulders, it was 65.3 ± 6.2 mm.

#### 3.2.3. Lateral Glenoidal Humeral Offset

The overall mean LGHO value was 48.9 ± 4.2 mm. The mean LGHO for the left shoulder joints was 48.6 ± 4.2 mm, while for the right shoulder joints, it was 49.1 ± 4.1 mm. In the younger patient group, the mean LGHO was the mean LGHO was 50.1 ± 4.5 mm, compared to 48.4 ± 3.9 mm in the older patient group. The mean LGHO for female shoulders was 46.4 ± 2.8 mm, whereas in male shoulders, it was 52.4 ± 3.1 mm.

#### 3.2.4. Humeral Shaft Axis Offset

The overall mean HAO was 25.2 ± 2.8 mm. The mean HAO for the left shoulder was 24.7 ± 2.8 mm, while for the right shoulder, it was 25.5 ± 2.8 mm. In the younger patient group, the mean HAO was 26.2 ± 3.2 mm, whereas in the older patient group, it was 24.8 ± 2.5 mm. The mean HAO for female shoulders was 24.0 ± 2.3 mm, and for male shoulders, it was 26.9 ± 2.6 mm.

#### 3.2.5. Cortical Offset

The overall mean CO was 15.7 ± 2.7 mm. The mean CO for the left shoulder was 16.0 ± 2.7 mm, while for the right shoulder, it was 15.6 ± 2.8 mm. In the younger patient group, the mean CO was 16.4 ± 2.9 mm, whereas in the older patient group, it was 15.5 ± 2.6 mm. The mean CO for female shoulders was 15.2 ± 2.7 mm, and for male shoulders, it was 16.5 ± 2.6 mm.

### 3.3. Subgroup Analysis

#### 3.3.1. Analysis Stratified by Side-Specific Differences for SOPs

Higher values of HO, GO, LGHO, and HAO were found on the right side. However, a significant difference (*p* < 0.05) was noted only for HAO between left shoulder joints (24.7 ± 2.8 mm) and right shoulder joints (25.5 ± 2.8 mm). No side-specific differences were found for HO (*p* = 0.139), GO (*p* = 0.567), LGHO (*p* = 0.567), or CO (*p* = 0.181). All the results are presented in Table 1. The details of the post hoc power analysis are shown in Appendix A.

#### 3.3.2. Analysis Stratified by Age-Specific Differences and Correlation Analysis of Age for SOPs

Significant differences (*p* < 0.05) were noted in all SOPs between the age groups. The younger patient (18–50 years) group exhibited higher measures in all SOPs. The mean HO in the older patient (over 50 years) group was 22.6 ± 2.3 mm, while in the younger patient group, it was 23.8 ± 2.5 mm. Similarly, higher values of GO, LGHO, HAO, and CO were noted in the younger patient group compared to the older patient group. All the results are presented in Table 2.

The correlation analysis revealed statistically significant (*p* < 0.001) inverse correlations between age and all SOPs, including HO (rho = −0.25), GO (rho = −0.20), LGHO (rho = −0.20), HAO (rho = −0.24), and CO (rho = −0.19). All the results of the correlation analysis are presented in Figure 2.

#### 3.3.3. Analysis Stratified by Gender-Specific Differences for SOPs

Males demonstrated significantly higher values in all SOPs compared to females (all *p* < 0.05). The mean humeral offset (HO) was 21.7 ± 2.0 mm in female patients, while a slightly higher mean HO of 24.6 ± 2.0 mm was observed in male patients. Additionally, male shoulder joints had a larger mean glenoid offset (GO) of 65.3 ± 6.2 mm compared to 60.1 ± 5.9 mm in females. Similarly, males exhibited higher mean values for LGHO (52.4 ± 3.1 mm), HAO (26.9 ± 2.6 mm), and CO (16.5 ± 2.6 mm) compared to females (LGHO: 46.4 ± 2.8 mm, HAO: 24.0 ± 2.3 mm, and CO: 15.2 ± 2.7 mm). All the results are presented in Table 3.

#### 3.3.4. Analysis Stratified by Grade of Osteoarthritis for SOPs

In our study, 467 patients were graded as Kellgren/Lawrence score 0–2 (SJH), and 33 patients were graded as Kellgren/Lawrence score 3–4 (SJD). No significant differences were noted between these two groups in any SOPs (HO: *p* = 0.259; GO: *p* = 0.226; LGHO: *p* = 0.864; HAO: *p* = 0.634; and CO: *p* = 0.799). All the results are presented in Table 4. The details of the post hoc power analysis are shown in Appendix A.

### 3.4. The Reliability of Measurement

The intra-observer reliability demonstrated ICCs ranging from 0.84 to 0.98, while the inter-observer reliability showed ICCs ranging from 0.93 to 0.99, reflecting excellent reliability (Appendix A). The results of the Bland–Altman plots are shown in Appendix A.

## 4. Discussion

The assessment of SOPs is crucial for diagnosing and treating shoulder pathologies [1,3,9,23]. Therefore, establishing precise radiological reference values for these parameters is essential to ensure the accuracy of diagnostic and therapeutic interventions [16,32]. To our knowledge, only a few articles describe the measurement method of SOPs and only one of them analyzes these parameters by assessing a large number of primary MRI images of the shoulder. In our study, we examined 500 MRI images of Asian shoulder joins to determine reference values for future applications in the diagnosis and treatment of shoulder pathologies.

The analysis stratified by side-specific differences showed higher values of HO, GO, LGHO, and HAO on the right shoulder compared to the left shoulder, but only HAO met the criterion of significant difference. The results mentioned above may be attributed to the prevalence of right-handedness in the study population, which accounts for around 90%, as most people predominantly use their right upper limbs [33]. Furthermore, there is a difference in the range of motion between the dominant and non-dominant shoulders [34,35]. Therefore, we suspect that the SOPs of different sides of the shoulder joint may be affected by handedness. To address this issue, further investigations focusing on the differences in SOPs among individuals with left- and right-handed dominance are essential. If applying this concept clinically, similar to our hip joint [36,37], relying solely on the physiological offset parameters of the contralateral shoulder joint for surgical interventions of severe degenerative diseases or fractures may not result in the reconstruction of the native morphology and physiology of the shoulder.

In our study, younger patients had significantly higher values of all SOPs. Furthermore, a weak negative correlation in all SOPs was noted. This may be attributed to the fact that peak bone mass and size are achieved during adolescence in women and later in men, and tend to decline with age [38]. Although we did not consider the potential influence of metabolic problems on bony morphology, the most reasonable explanation for lower SOPs in aging is age-related bone loss.

In addition, the analysis stratified by the grade of osteoarthritis showed that all measures of SOPs were not influenced by the stage of arthritis. From our perspective, osteoarthritis is associated with higher bone density and the abnormal growth of bone, which is known as osteophytes. Due to the variable morphology of osteoarthritis in the humeral head, the location of osteophytes may influence the value of SOPs. Thus, there is no significant difference in all SOPs between higher (KL 3–4) and lower (KL 0–2) grades of osteoarthritis. Nevertheless, a previous study by Knowles et al. disclosed that the articular diameter of the humeral head in individuals with osteoarthritis was notably larger compared to the diameter of a normal humeral head [39]. The plausible reason is that, in our study, only 6.6% of patients belonged to the SJD group (KL 3–4). Therefore, the difference in size between the two groups may greatly influence the result. Future studies assessing osteoarthritis-dependent analysis with equal sizes of SJH and SJD groups may significantly improve our understanding of the progression of osteoarthritis and its relation with offset parameters.

In our study, significantly higher measurements for all SOPs in male patients compared to female patients were noted. Several studies have already revealed that certain parameter offsets of shoulder morphology are related to gender. For example, Derrick et al. conducted an assessment of the glenoid dimensions of 993 cadaveric specimens, revealing a greater glenoid width in males [40]. Similarly, Heri et al. reported larger humeral heads and glenoids in males than in females by analyzing 156 shoulder CT scans with 3D reconstruction techniques to measure the morphometry of the humeral head and glenoid [41]. Our research not only demonstrated that the SOPs are different between genders but also organized the values of SOPs for male and female groups.

To establish a reliable approach for reproducible measurement of SOPs, precise morphological landmarks and standardized image examination are necessary. However, in our clinical experience, the variability in the quality of radiological imaging makes the identification of landmarks difficult. Some studies have already explored the issue of establishing methods to record morphological parameters in the shoulder joint [9,19,42,43]. To the best of our knowledge, only Meier has described how to measure SOPs by interpreting MRI images of shoulders. Moreover, they also developed a new measurement method to record GO, which can be easily applied in daily clinical scenarios [23]. Given the excellent intra-observer and inter-observer reliability in this study, the methods we used to measure SOPs were replicable.

Humeral offset and glenoidal offset are crucial factors in the diagnosis and treatment of shoulder diseases. To our knowledge, only a few articles describe the measurement method of SOPs and only one of them analyzes these parameters by assessing a large number of primary MRI images of the shoulder. Our study examined 500 MRI images of Asian shoulder joints to determine reference values for future applications in the diagnosis and treatment of shoulder pathologies, especially in the Asian group. We established a valid reference for SOPs in the Asian population, which should be essential to consider in routine practice. While this study focuses solely on radiological measurements without addressing clinical outcomes, it nonetheless provides valuable insights for decision-making in the treatment of shoulder disorders across different age and sex groups. Additionally, the findings can serve as a useful reference for future implant designs specifically tailored to the Asian population.

However, this study still had several limitations. First, we did not compare the results between MRI study and other radiological examinations, such as X-rays or CT scans. Although our study focused more on anatomical bony parameters, we still cannot avoid the drawbacks of MRI such as oversensitivity in diagnosis of soft tissue injury. Secondly, we had no idea about the influences of metabolic bone diseases, such as vitamin D deficiency and osteoporosis, on normal SOPs, since our study lacked the information of clinical data. Third, the number of patients with high-grade osteoarthritis stage is also small, we had no comments on the effect of advanced shoulder osteoarthritis on SOPs. Last but not least, we did not investigate whether handedness affects the measures of bilateral joints. Further studies with records of handedness may be needed to solve this dilemma.

## 5. Conclusions

In this study, we found significantly higher values for all types of SOPs in males and younger patients, compared to their counterparts. Additionally, the right shoulders did not have higher mean values of SOPs than the left shoulders, except HAO, which indicates that using the contralateral side of the joint as a reference for surgical planning seems reliable. The aforementioned results should be considered when planning shoulder surgery.

## Figures and Tables

**Figure 1 diagnostics-15-00146-f001:**
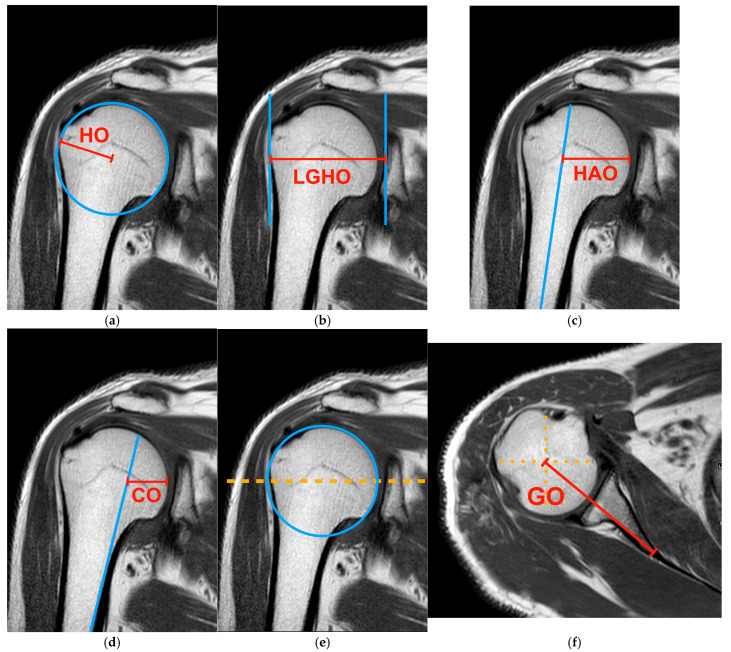
HO, LGHO, HAO, and CO measurements were performed in the coronal view of the shoulder joint on MRI scans, while GO was evaluated in the axial view. (**a**) To assess the HO, the central point of rotation of the humeral head was identified, and the distance from greater tuberosity to this point was measured as the HO. (**b**) To measure the LGHO, we determined the center point of the glenoid fossa. The distance from greater tuberosity to this point is LGHO. (**c**) HAO was determined by measuring the distance from the humeral head shaft axis to the medial cortical edge of the humeral head. This was achieved by drawing a horizontal line that intersects the central point of the humeral head. (**d**) To measure the CO, we used the same horizontal line as for the HAO and identified the axis of the medial cortical bone of the humerus. The CO is defined as the distance from this axis to the medial cortical edge of the humeral head. (**e**) Measuring the GO requires first identifying the center of rotation of the humeral head in the coronal view. (**f**) Then, we used measurement software to locate the corresponding point in the axial view. GO was defined as the distance between this point and the neck of the scapula.

**Figure 2 diagnostics-15-00146-f002:**
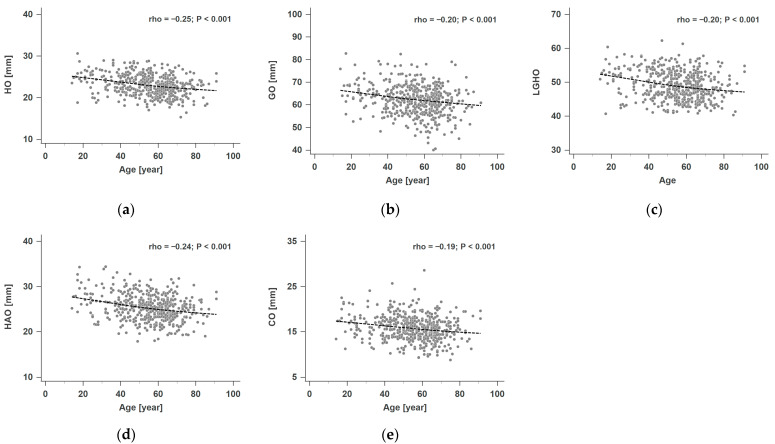
Correlation analysis between patients’ age and HO, GO, LGHO, HAO, and CO. The analysis showed a significant correlation (*p* < 0.001) between patients age and (**a**) HO (rho = −0.25), (**b**) GO (rho = −0.20), (**c**) LGHO (rho = −0.20), (**d**) HAO (rho = −0.24), and (**e**) CO (rho = −0.19).

**Table 1 diagnostics-15-00146-t001:** Analysis of side-specific differences between offset parameters.

	Total (*n* = 500)	Left (*n* = 200)	Right (*n* = 300)	*p*-Value
HO (mm)	22.9 ± 2.4	22.8 ± 2.6	23.0 ± 2.4	0.139
GO (mm)	62.3 ± 6.6	62.2 ± 6.6	62.3 ± 6.6	0.567
LGHO (mm)	48.9 ± 4.2	48.6 ± 4.2	49.1 ± 4.1	0.567
HAO (mm)	25.2 ± 2.8	24.7 ± 2.8	25.5 ± 2.8	0.011 *
CO (mm)	15.7 ± 2.7	15.9 ± 2.7	15.6 ± 2.8	0.181

* *p* < 0.05.

**Table 2 diagnostics-15-00146-t002:** Analysis of age-specific (age in years) differences between offset parameters.

	Total (*n* = 500)	18–50 y (*n* = 155)	>50 y (*n* = 345)	*p*-Value
HO (mm)	22.9 ± 2.4	23.8 ± 2.5	22.6 ± 2.3	<0.001 *
GO (mm)	62.3 ± 6.6	64.1 ± 6.3	61.5 ± 6.5	<0.001 *
LGHO (mm)	48.9 ± 4.2	50.1 ± 4.5	48.4 ± 3.9	<0.001 *
HAO (mm)	25.2 ± 2.8	26.1 ± 3.2	24.8 ± 2.5	<0.001 *
CO (mm)	15.7 ± 2.7	16.6 ± 2.9	15.5 ± 2.6	0.002 *

* *p* < 0.05.

**Table 3 diagnostics-15-00146-t003:** Analysis of gender-specific differences between offset parameters.

	Total (*n* = 500)	Females (*n* = 289)	Males (*n* = 211)	*p*-Value
HO (mm)	22.9 ± 2.4	21.7 ± 2.0	24.6 ± 2.0	<0.001 *
GO (mm)	62.3 ± 6.6	60.1 ± 5.9	65.3 ± 6.2	<0.001 *
LGHO (mm)	48.9 ± 4.2	46.4 ± 2.8	52.4 ± 3.1	<0.001 *
HAO (mm)	25.2 ± 2.8	24.0 ± 2.3	26.9 ± 2.6	<0.001 *
CO (mm)	15.7 ± 2.7	15.2 ± 2.7	16.5 ± 2.6	0.002 *

* *p* < 0.05.

**Table 4 diagnostics-15-00146-t004:** Analysis of osteoarthritis-specific differences between offset parameters.

	Total (*n* = 500)	KL 0–2 (*n* = 467)	KL 3–4 (*n* = 33)	*p*-Value
HO (mm)	22.9 ± 2.4	23.0 ± 2.4	22.3 ± 2.6	0.259
GO (mm)	62.3 ± 6.6	62.4 ± 6.6	60.7 ± 6.5	0.226
LGHO (mm)	48.9 ± 4.2	48.9 ± 4.2	48.9 ± 4.0	0.864
HAO (mm)	25.2 ± 2.8	25.2 ± 2.8	25.4 ± 3.2	0.634
CO (mm)	15.7 ± 2.7	15.8 ± 2.7	15.7 ± 3.1	0.799

KL, Kellgren/Lawrence grade.

## Data Availability

The raw data supporting the conclusions of this article will be made available by the authors upon request.

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
