# Peer review of "Shoulder Physiological Offset Parameters in Asian Populations—A Magnetic Resonance Imaging Study"

_diagnostics, 2025, doi:10.3390/diagnostics15020146_

Round 1

Reviewer 1 Report

Comments and Suggestions for Authors

This is a very interesting and well executed study that will be of use to both clinicians and researchers in anatomical sciences. There are some shortcomings here and there that I believe the authors should address before this paper can be published.

1.In elaborating on the importance of shoulder offset parameters (SOPs) for the diagnosis and treatment of shoulder disorders, additional clinical data or research examples could be further cited to more fully illustrate the urgency of establishing physiological reference values for the shoulder in Asian populations.

2. Although humeral offset (HO) and glenoidal offset (GO), lateral glenoid humeral offset (LGHO), humeral shaft axis offset (HAO) and cortical offset (CO) have been defined previously in the literature, perhaps a brief elaboration in this paper would be beneficial to the reader's understanding of the paper. Furthermore, Please describe in further detail how the selection of images for localisation of the centre of rotation of the humeral head was made from a large number of coronal images

3. We agree with the study population as categorised by the authors into a younger patient group and an older patient group. However, there is a relative lack of discussion and evidence as to why 50 years of age was defined as a grouping criterion for the younger and older patient groups. We suggest that the authors read the relevant article (PMID: 36579520; PMID: 16337525) and then provide a brief explanation in the manuscript.

4. The overall language of the thesis is more fluent, but some statements can be further optimised to improve the readability of the thesis. For example, some long sentences could be split appropriately, and please provide the full name of an acronym when it first appears.

Reviewer 2 Report

Comments and Suggestions for Authors

Dear authors,

Thank you for the opportunity to revise this paper and provide comments.

This study aims to establish physiological reference values of shoulder parameters for Asian populations and to 79 investigate side-, age-, and gender-specific differences in SOPs, and explore their relationship with OA grades.

This study is interesting and has been conducted appropriately to my understanding. However, I think that there as some issues regarding the statistical procedures.

At first, you need to estimate a priori the minimum sample needed for this study to ensure power adequacy. I think that the GPower software could be used and provide in the text the procedures used to estimate the minimum sample size with the GPower. Also, maybe you can re-check that the sample that you actually recruited was adequate using this sample again in the GPower and the results of your study.

More importantly, you have not stated the minimum requirements for running the statistical tests that you mentioned. Why did you use Spearman’s? Did you have problems with the normal distribution? Did you check for normal distribution of your data and how?

As for the reliability of the measurement, you can also conduct a Bland and Altman and show the plot in the text. This would clearly help the readers to visualize agreement and data spread.

Lastly, you could also check whether an association exists s between different SOPs and other demographic characteristics.

I hope this would help you improve the scientific soundness of the manuscript. I am happy to review this manuscript again after revisions.  
